# Is the Thoughts and Health programme feasible in the context of Swedish schools? A quasi-experimental controlled trial study protocol

Carl Wikberg [ID] ,[1,2] Pia Augustsson [ID] ,[1,2] Gudny Sveinsdottir,[2]
W Edward Craighead [ID] ,[3] Erikur Örn Arnarson,[4,5] Ina Marteinsdottir,[6]
Josefine L Lilja [ID] [7]

For numbered affiliations see end of article.

**Correspondence to**
Dr Carl Wikberg;
carl.wikberg@allmed.gu.se

## ABSTRACT

**Introduction** Clinical depression is a substantial problem among adolescents, increasing significantly at about age 15 years. It causes impairment in social, academic and familial relationships, as well as ongoing cognitive and emotional difficulties for the individual. A study in Iceland demonstrated that a cognitive–behavioural, developmentally based intervention programme, 'Thoughts and Health', prevented initial episodes of depression and/or dysthymia (DYS) (major depressive disorder/DYS) in adolescents for up to 12 months following completion of the programme. We would like to test the feasibility of implementing the Icelandic method in a Swedish context and to evaluate the long-term effects of such a programme.

**Methods and analysis** A quasi-experimental controlled design, combined with qualitative and quantitative methods, will be used to address the research questions. In this study, 617 children aged ~14 years will be screened for depression, and those "at risk" for development of clinical depression will be offered a 12 week course, 'Thoughts and Health'. This course aims to prevent first depression in adolescents. A comparable group of children will function as controls.

Depending on the type of variable, baseline comparisons between the two groups of relevant initial measures will be evaluated with t-tests or $\chi^2$ analyses. The effects of the programme on the development of clinical levels of depression will be evaluated using the follow-up data of 6, 12 and 18 months. Index parental depression at baseline will be tested as a moderator in the evaluation of the effects of the prevention programme.

**Ethics and dissemination** This study is approved by the Swedish Ethical Review Board (reference number 2019–03347) in Gothenburg.

We plan to disseminate the knowledge gained from this study by publishing our results in peer-reviewed scientific journals and other scholarly outlets.

**Trial registration number** NCT04128644; Pre-results.

## INTRODUCTION

Clinical depression is a substantial problem among adolescents, increasing significantly at about the age of 15 years. It causes impairment

### Strengths and limitations of this study

► Quasi-experimental controlled design will provide information about whether the intervention prevents depressive episodes in adolescence in a Swedish sample.

► Qualitative data will shed light on how the intervention was perceived by the adolescents and how the personnel experienced the implementation process.

► Quantitative data will provide information regarding adolescents' mental health and social functioning.

► Follow-up assessments will provide information regarding the intervention's long-term effects on school performance and mental health.

► The study's lack of a randomised controlled design and the few participants limit its generalisability.

in social, academic and familial relationships, as well as ongoing cognitive and emotional difficulties for the individual.[1 2]

A study in Iceland demonstrated that a cognitive–behavioural, developmentally based intervention programme called 'Thoughts and Health' prevented initial episodes of depression and/or dysthymia (DYS) (major depressive disorder (MDD)/DYS) among Icelandic adolescents for up to 12 months following completion of the programme.[3 4] Other studies have demonstrated that similar programmes can be employed to prevent depression among teenagers who have never been depressed and/or who previously have been depressed.[5–8] The Thoughts and Health programme uses principles of both behavioural (eg, relaxation and behavioural activation (assertion and social approach)) and cognitive (eg, identification and modification of irrational thoughts, thinking errors and self-beliefs) interventions. The programme procedures were specifically adapted to the appropriate

developmental level of the youth participants, with developmental level defined as stage 5 (identity vs confusion) of Erikson's conceptual model of human psychosocial development.[9] Specifically, stage 5 encompasses adolescence, during which individuals *develop a sense of self* within the context of emerging independence from the family and expansion of interpersonal/social relationships.

However, despite the positive impact of such programmes, preventive effects have not been consistently obtained.[10–12] Reviews have concluded that universal prevention programmes (programmes applied to the general population) have not been effective in preventing depression[13 14], and excellently conducted studies in Australia have provided data consistent with that conclusion.[15 16] In general, only targeted programmes (*selective* programmes—for populations exposed to risk factors, or *indicated* programmes—for populations who are experiencing some symptoms) have shown significant short-term effects in preventing depression.[1 5–7]

Most programmes aimed at preventing depression have included both youth who have suffered a prior depression and those who have never been depressed. The study of prevention of relapse or recurrence in prevention programmes for those who have already suffered an MDD is clearly useful, but earlier studies have not distinguished between prevention of the first episode of MDD and the prevention of relapse/recurrence of MDD. However, because of the recurrent nature of the disorder and the differing causal factors and probability patterns of first versus subsequent episodes[17] and in order to prevent the deleterious effects of an episode of MDD on youth, it is especially important to study the prevention of the first or initial episode among never previously depressed subjects.

Since clinical depression among adolescents increases significantly at about the age of 15 years, and Swedish children finish their compulsory school (junior high school) at that age, being at risk of depression can be a reason for failing their final examinations, thereby limiting their chances for further academic studies.

We would like to test the feasibility of implementing the Icelandic method in a Swedish context and to evaluate the long-term effects of such a programme. We would like to test the method with a quasi-experimental controlled trial and to evaluate the outcomes after 6, 12 and 18 months. The control group will be assessment-only adolescents; they receive ordinary 'Student Health'; they participate in all study assessments, but they are not offered the intervention Thoughts and Health.

This is, to our knowledge, the first study in Sweden that employs a longitudinal design to evaluate the effects of a school-based programme aimed at preventing depression in adolescents.

This study will be implemented together and within the municipalities of Tjörn and Orust, two islands off the west coast of Sweden, with each island having only two middle schools. Thus, this study is built on a close collaboration between academia, primary care and the community.

We are hopeful that this programme will be transferrable to a Swedish context and that it will be possible to further develop and enhance the method. Although Iceland and Sweden are similar in many ways, the school system and cultural context differ to some extent.

We hypothesise that programme implementation will reduce the risk of developing depressive symptoms among participating students compared with their control counterparts, and we will examine this in participating adolescents up to 18 months after programme completion. We also hypothesise that adolescents in the intervention group who participate in the programme will finish school with complete grades to a greater extent (more students complete their schooling) than the control group.

## AIM

This study aimed to test the feasibility of implementing an Icelandic cognitive–behavioural programme designed to prevent depression, Thoughts and Health,[3 4] in a Swedish school setting.

The perspectives of both the participating adolescents and the primary care professionals who lead and support the programme in the schools will be taken into account when evaluating the programme's feasibility.

The study also aimed to evaluate the number of students who finish junior high school (ninth grade, ~15 years old) with complete grades. In this study, the outcomes concerning complete grades of the students participating in the intervention programme will be compared with the complete grades of those in the control group.

Our primary research questions are

Is the Thoughts and Health programme feasible in the context of Swedish schools, from the perspective of the students participating in the programme?

Is the Thoughts and Health programme feasible in the context of Swedish schools, from the perspective of the primary care professionals leading and supporting the programme implementation?

Our secondary research questions are

Does participation in the Thoughts and Health programme affect the number of students developing depressive symptoms?

Will participation in the course Thoughts and Health increase the participants' chances of completing compulsory school with high school eligibility?

## METHOD
### Study design
A quasi-experimental controlled design combined with qualitative and quantitative methods will be used to address the research questions (Clinicaltrials.com).

We have used the SPIRIT reporting guidelines.[18]

### Participants
An invitation regarding participation in the study to high schools in the area resulted in four junior high schools from the municipalities of Tjörn (Häggvall and Bleket,

intervention arm) and Orust (Henån och Ängås, control arm) that will participate in the study. The allocation of the schools was chosen on the basis of their rural location and socioeconomic similarity, willingness to participate and available staff. Since all schools had shown interest in participating in this study, the assignment of intervention and control arms was determined on the basis of available resources; therefore, Tjörn was chosen for the intervention programme at this stage. Two intervention courses were planned (I and II), with course I beginning at the start of the fall semester (~1 September) 2019 and course II beginning at the start of the fall semester (~1 September) 2020.

In total, approximately 320 eighth grade Swedish adolescents aged 14 years will participate in a preintervention screening process per course. In the preintervention screening process, all students will complete the Children's Depression Inventory (CDI, described further)[19] assessment instrument via an online questionnaire tool. This will be part of the existing "Student Health" ordinary assignments that include health and mental health status checkups as per the Swedish School Law.

We are aware of the small number of students that will be part of the collected data at this early stage, and we are planning a full-scale randomised trial once we have completed this first study; hence, there were no power calculations of sample size at this stage.

All students who score at or above the 75th percentile on the CDI screening will be qualified initially for possible inclusion in the study (expected n=80 total per course). They will receive an invitation together with their parents to attend a psychological assessment interview (the Mini International Neuropsychiatric Interview for Children and Adolescents (MINI-KID), described further).[20] At the interview, they will be given the opportunity to discuss their CDI result and if needed, to be channelled to the right level of care (eg, if a student scored very high on CDI, the psychologist will need to decide whether there is any underlying diagnosis that requires treatment or if the student has any other condition that also requires immediate attention and treatment). If a student is diagnosed with depression at this phase, he or she will no longer be eligible to participate in the study and will be directed to appropriate mental healthcare resources in the community. Prior to attending this psychological assessment, students and their parents will separately fill out the Revised Children's Anxiety and Depression Scale (RCADS)[21] assessment instrument.

Based on the MINI-KID results, students who meet criteria for having any current or past major depression, DYS, bipolar disorder I or II, cyclothymia, anorexia, bulimia, psychotic disorder, alcohol or substance dependence, attention deficit hyperactivity disorder, oppositional defiant disorder or conduct disorder will be excluded. All other students thus fulfil the inclusion criteria. All students, both in the intervention and control arms, will be invited to participate in the study via an appropriately worded letter (expected n=40 total

per study arm). See table 1 for inclusion and exclusion criteria.

Thus, about 20 students per course will be offered to participate in the intervention programme. These students will be approached by the programme staff and offered the opportunity to participate in the baseline clinical interview and assessment, the intervention programme, and postintervention follow-up assessments at 6, 12 and 18 months after programme completion (figure 1).

Participants in the intervention and control arm will be offered tickets to the cinema as a gesture of gratitude for their participation in each of the follow-up assessments.

## Measures and assessments
Measures and assessments are described as follows. The instruments, persons involved and the time point for each measure or assessment are shown in table 2. All psychometric measures are administered by esMaker (htpps://entergate.SE/products/esmaker/). Course surveys are distributed in the form of hard copies. If participant frequency is shown to be low, those who have not completed the measures will receive a reminder by email.

## Outcomes
### CDI, ordinal variables (collected during preintervention phase, at course start, and at 6, 12 and 18 months of follow-up)
The CDI is a 27-item self-reported, symptom-oriented scale that rates the severity of depression during the past month. It is designed for children/adolescents, ages 7–17 years. Each of the 27 items comprises three sentences used by the student to rate the severity of the symptoms. This inventory has previously shown excellent internal

**Table 1** Inclusion and exclusion criteria

| Inclusion criteria | Exclusion criteria |
|---|---|
| Eighth grader | Ongoing or past depression |
| Guardian of eighth grader | Addiction |
| Understand Swedish language skills, written and oral | Past or current suicidal attempt or ideation |
| Be able to fill out questionnaires independently | Neuropsychiatric diagnosis that precludes group activities |
| Completed CDI and be above 75th percentile | Not high enough level of function on the CGAS (cut-off≥61) |
| Completed psychological assessment | Lacking sufficient Swedish language skills, written and oral |
| Written and oral consent | Serious mental illness |
| | Ongoing other psychotherapeutic treatment |

*Assessed and determined by a psychological assessment, based on the clinical rating instruments CDI, MINI-KID, RCADS and the children's current status.
CDI, Children's Depression Inventory; CGAS, Children's Global Assessment Scale; MINI-KID, Mini International Neuropsychiatric Interview for Children and Adolescents.

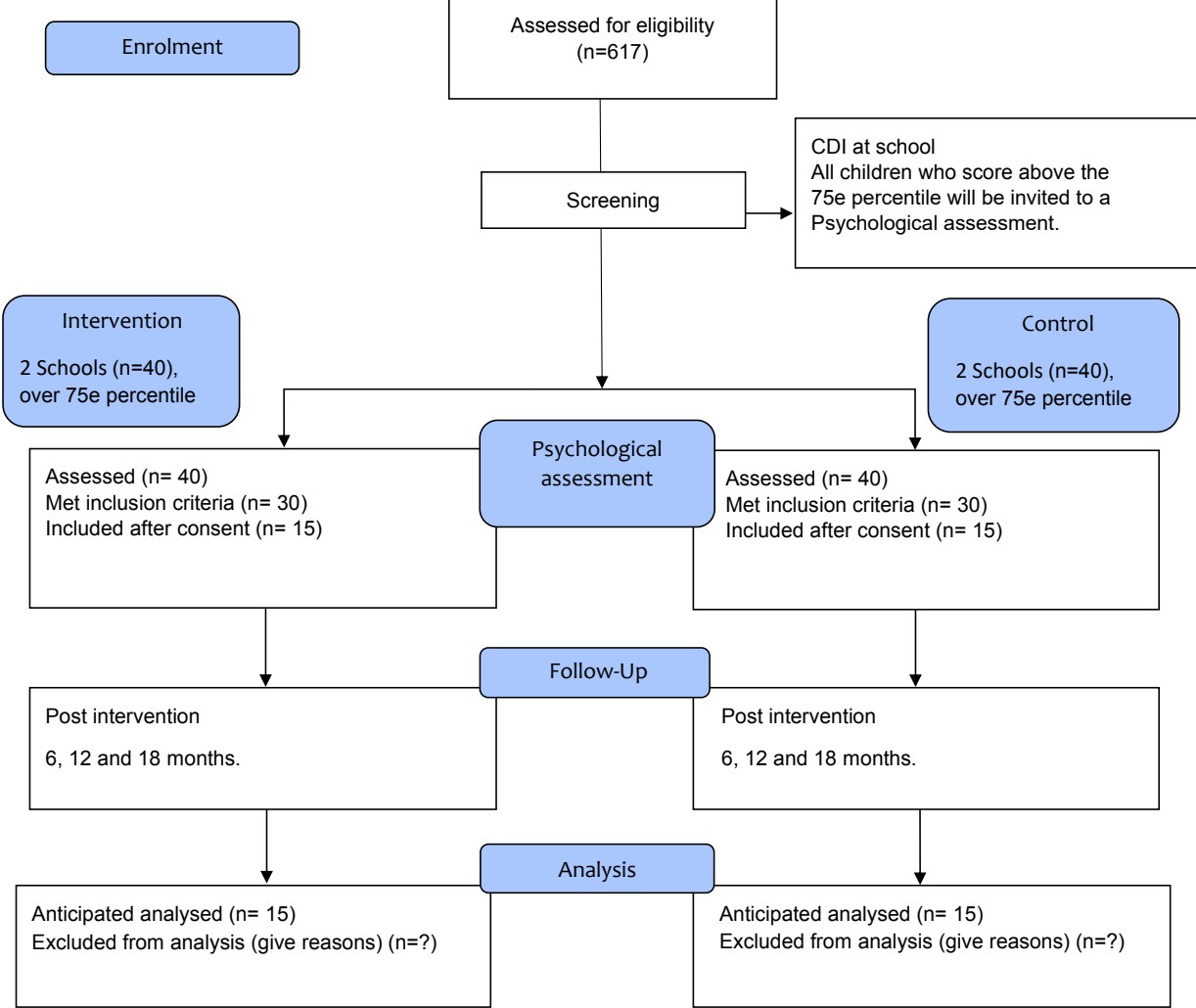

**Figure 1** Flowchart over the Thoughts and Health study.

consistency (Cronbach alpha coefficients of 0.83–0.94). The present study uses the total score of the Swedish version of the CDI.[19] The CDI will be used both in screening and in follow-ups with participating children.

### RCADS, ordinal variables (collected during preintervention phase and at 6, 12 and 18 months of follow-up)

RCADS is a widely used questionnaire designed to assess clinical syndromes of anxiety as well as depression, based on the *Diagnostic and Statistical Manual of Mental Disorders* (DSM), Fourth Edition, criteria. The RCADS provides two total scores and has six subscales: separation anxiety disorder, social phobia, obsessive–compulsive disorder, panic disorder, generalised anxiety disorder (GAD) and MDD. The internal consistency of the RCADS subscales is high, with Cronbach alphas ranging from 0.78 to 0.88. There are two versions of RCADS: one for adolescents (RCADS) and one for parents (RCADS-P).

### MINI-KID, categorical variables (collected during preintervention phase and at 6, 12 and 18 months of follow-up)

The MINI-KID is a structured interview for children and young people aged 4–17 years old.[20] It is based on

diagnoses using the DSM, Fifth Edition,[22] and the 10th edition of International Statistical Classification of Diseases and Related Health Problems. Both parents and youths will participate in the interview, but the interview can also be conducted with only young people. The interview aims to ensure diagnosis and inclusion/exclusion criteria.

### Demographic information and somatic health, categorical variables (collected at course start and at 6, 12 and 18 months of follow-up)

The baseline questionnaire will include items regarding sociodemographics, physical activity, drug use, and some questions about mobile and internet usage.

### Children's Global Assessment Scale (CGAS) ordinal variables (collected during preintervention phase)

The CGAS will be used to measure global functioning.[23] The assessment will be made by the person who conducts the MINI-KID interview. The CGAS estimate is made on a scale of 1–100, where a higher score indicates a higher functional level. This will be completed for participating children.

**Table 2** Instruments and follow-up assessments

| | Screening | Pre intervention | During treatment 6 weeks | Post-treatment 3 months | 6 month follow-up | 12 month follow-up | 18 month follow-up |
|---|---|---|---|---|---|---|---|
| **Clinician-rated instruments** | | | | | | | |
| MINI-KID | | x | | x | x | x | x |
| CGAS | | x | | x | x | x | x |
| Clinical Global Impressions (CGI) | | | | x | x | x | x |
| **Self-rated instruments** | | | | | | | |
| CDI | x | x | x | x | x | x | x |
| RCADS | x, o | x | x | x | x | x | x |
| EWSAS | | x | x | x | x | x | x |
| MHC-SF | | x | x | x | x | x | x |
| Baseline Questionnaire | | x, o | | x, o | x, o | x, o | x, o |
| **Course surveys** | | | x | x | x | x | x |
| **Qualitative method** | | | | 3 months | | | |
| Focus group discussions | | | | x | | | |
| Individual interviews | | | | x | | | |

Parents: o, adolescents: x.
CDI, Children's Depression Inventory; EWSAS, Education Work and Social Adjustment Scale; MHC-SF, Mental Health Continuum–Short Form; RCADS, Revised Children's Anxiety and Depression Scale.

## Education Work and Social Adjustment Scale (EWSAS) ordinal variables (collected at course start, course middle (6 weeks after course start), and at 6, 12 and 18 months of follow-up)

The EWSAS measures general level of function.[24] It is an adaptation of the Work and Social Adjustment Scale and measures the level of function in five areas: relationships, family, leisure time, social activities, and school and everyday activities. This will be completed by participating children.

## Mental Health Continuum–Short Form (MHC-SF), ordinal variables (collected at course start, course middle (6 weeks after course start), and at 6, 12 and 18 months of follow-up)

The MHC-SF is an increasingly used questionnaire that measures three components of well-being: emotional, social and psychological. It has shown excellent internal consistency (>0.80) and discriminant validity in adolescents (ages 12–18 years) and adults in the USA, in the Netherlands and in South Africa.[25–28] MHC-SF consists of 14 items asking about the frequency of well-being experiences during the last month. There are three items concerning emotional well-being, such as feelings of being happy, interested in life and satisfied, five items concerning social well-being, reference, actualisation, contribution, coherence and integration, and six items concerning psychological well-being capturing purpose in life, environmental mastery, autonomy, personal growth, positive relations and self-acceptance. Answers are given on a 6-point Likert scale from 0=never, 1=once or twice, 2=about once a week, 3=about two or three times a week,

4=almost every day and 5=every hour. Cronbach's alpha was 0.92 for emotional well-being, 0.88 for social well-being and 0.92 for psychological well-being. Scores on each well-being component are summed and categorised as 'flourishing', 'moderately healthy' or 'languishing' according to official MHC-SF guidelines. The MHC-SF has been useful in improving the scientific understanding of the risk of future mental illness (major depressive episode, GAD, panic attacks, risk of premature mortality, healthcare use, missed days of work, suicidality and self-reported academic impairment among college students). The MHC-SF will be completed by participating children.

## Parental depression, binary variable (collected during preintervention phase)

At the time of the baseline interview, adolescents will be asked if their parents have previously suffered from or currently suffer from depression. Similarly, parents will be asked if they have previously suffered from or currently suffer from depression. These data are collected because there are known associations between depression in parents and the health of their children.[6] We would like to know if any of the parents have had or currently have a depressive episode as that may be related to the depressive outcomes in their children. Both adolescent and parent data will be analysed separately. This will be useful for further analysis regarding long-term follow ups. As an example, we will be able to see whether intervention group and control group parents are similar regarding possible previous or current state of depression.

### Course surveys, ordinal variables (collected after each course session)

After each course session (total of 12 sessions per course), the students and the primary care professionals leading and supporting the course will directly provide their feedback about the course session by answering a short survey. Different surveys will be provided to students and the professionals, respectively. These surveys will assess feasibility, satisfaction and overall impression of the session.

### Course attendance and dropouts, binary variables (collected during each course session)

Attendance and dropouts will be recorded at each course meeting, as a measure of the feasibility and acceptance of the programme.

### Focus groups and interviews, qualitative analysis (collected within 1 month after course end and at 18 months follow-up)

All participants (students and professionals) will be invited to take part in focus group discussions or individual interviews after the completion of the full course. These discussions will be held separately, with students in one or two groups and the professionals in another, or as individual interviews. Focus groups will be held with the students and individual interviews with professionals.

### Final junior high school grades, categorical variables (collected at approximately 18 months after course end)

Students' final grades in junior high school (ie, in ninth grade, or 18 months after their inclusion in the programmes) will be collected to be able to evaluate the number of students who finish school with complete grades. The information will be provided by the school administration on consent (online supplemental file) from the students.

All psychological assessments will be conducted by a trained psychologist from the primary care centre, and the focus groups and interviews will be conducted by the authors CW (PhD and specialist nurse) and PA (PhD student).

### Data management

A research journal will be used to follow up the person during the programme and follow-up period. The research journal will be kept at the primary care research and development unit, Kungsgatan, in Gothenburg, where data from the research journal are entered. Data in the database at the research unit will only be entered with code numbers. Only the research leaders will then have access to the code key, which is stored in in a locked safe at the research unit. The research journals will be kept at the research unit and will be anonymised during the evaluation phase. The composition of the research group may change during the course of the project, but only authorised persons who are part of the research group will have access to data. The usual privacy is maintained.

### Intervention: prevention programme

The prevention programme will be based on the Icelandic Thoughts and Feelings programme[1] and will include 12 group sessions of 90 min duration, delivered two times per week for 2 weeks and then once a week for 8 weeks in the school setting by Student Health school psychologists. School nurses will also attend each session as support for the students. The programme uses two intervention manuals: a group leader manual and a student manual. Both of these manuals have been translated first from Icelandic to English, and then from the English version into Swedish, and finally corrected for language in the Swedish version. After translation, the manuals were additionally reviewed by several eighth-grade students for cultural acceptability (eg, word choice) and adapted as per their feedback. The course material is based on cognitive behaviour theory, and the course will be held as a group activity. The course was originally developed as a group activity, and we will test this method in our study, as part of the evaluation of whether the method is suitable for Swedish adolescents of today. The course contains group sessions focusing on a specific topic, after which they will describe their state of health and feelings in a diary and receive some home assignments to be completed between sessions. All sessions start with a light refreshment and a summary of the previous session. Session #1 is an introduction. Session #2 is about 'the spiral', that is, how negative or positive thoughts and actions affect subsequent actions and feelings. Session #3 is about relaxation. Session #4 will discuss the concept of a 'baseline'. The remaining sessions deal with the following topics: setting targets (session #5), feelings (session #6), negative thoughts (session #7), irrational thoughts (session #8), self-perception (session #9), communication (session #10), stop and reconsideration (session #11) and course consummation (session #12, final session).

### Statistical analysis

Depending on type of variable, baseline comparisons of the two groups of relevant initial measures will be evaluated with t-tests or $\chi^2$ analyses. The effects of the programme on the development of depressive symptoms will be evaluated for the data of 6, 12 and 18 months using the Cox proportional hazards model to estimate survival curves and rates for different thresholds of combined scores of depressive symptoms. Index parental depression at baseline will be tested as a moderator in the evaluation of the prevention programme effects. Even though the sample in this part of the project is rather small, we will stratify all results by selected measures collected in the baseline questionnaire including gender, socioeconomic status and mobile/internet use, among others. With regard to the small sample size in this early stage of the study, dropouts or non-completion of outcome measures will be analysed with intent-to-treat.

The qualitative data collected through interviews and focus group discussions will not require any statistical

analysis, but they will be analysed using thematic qualitative analytical methods.

## Ethics and dissemination

This study is approved by the Swedish Ethical Review Board (reference number 2019–03347) in Gothenburg. Since participants are adolescents aged <15 years, consent from both parents (all parents if eligible) and the adolescents themselves will be obtained, as required by applicable laws and regulations.

This study is based on a validated programme that has been conducted in other countries showing good preventive effect. Based on previous studies, we expect preventive effects of the intervention in the participating adolescents. To ensure that the risks are minimal throughout the programme implementation and follow-up, we will provide all participants from the preintervention phase through postintervention follow-up, in both control and intervention arms, with access to professional help in the form of psychologists, doctors and nurses if needed. The research group has solid experience implementing the treatment programme in other settings; their experience indicates that the intervention is perceived as positive by students and that participation in the group work has not been stigmatising. All measures and assessments used in the study are validated and tested in many other studies. All researchers and professionals in this study have good paediatric psychiatric competence. They are accustomed to talking with adolescents and will be able to carefully and sensitively handle unpredictable situations. All personnel involved in the study undergo training in the intervention. The research team is trained in good clinical practice (GCP) (scientific quality and ethical standards for planning, conducting, recording and reporting trials involving human participation). The study subjects will never be alone during the intervention because the intervention is carried out in groups and by trained staff. We will ensure the privacy of study participants, and all relevant legislation will be followed regarding use of research data. All members of the research group have undergone training in the intervention by the founders of the programme. The research group consists of general practitioners, psychologists, specialist nurses and social workers. The trial manager is a specialist psychologist and a PhD, and the implementation leader is a specialist nurse and PhD. The founders of the programme are both professors of psychology, one in the USA and the other one in Iceland.

Results from the study will be presented so that no individuals, including students, parents and the employer representatives, will be identifiable. All results will be presented on a group level. All participation is voluntary, and the student can withdraw from the study at any time.

The trial will be conducted in compliance with this study protocol, the Declaration of Helsinki and its modifications, and GCP.

We plan to disseminate the knowledge gained from this study by publishing our results in peer-reviewed scientific journals, giving presentations at national and international conferences, and spreading information through local and national media channels. We will approach local and national politicians to ensure prolongment of the project.

## DISCUSSION

The study protocol of this research project shows that the planning and organisation of the research make it feasible to answer the four research questions.

Question (1). Is the Thoughts and Health programme feasible in the context of Swedish schools, from the perspective of the students participating in the programme? The answer will be based on the qualitative interviews and focus group discussions conducted with adolescents participating in the intervention group, as well as analysis of the course surveys gathered after each session and analysis of possible dropouts.

Question (2). Is the Thoughts and Health programme feasible in the context of Swedish schools, from the perspective of the primary care professionals leading and supporting the programme implementation? The answer will be based on the analysis of interviews with the professionals who have taken part in individual interviews after the completion of the intervention course, as well as analysis of the course surveys gathered after each session.

Question (3). Does participation in the Thoughts and Health programme have an effect on the development of depressive symptoms? The answer will be based on collected data and statistical analyses. Outcomes will be collected before, at and after baseline, based on adolescents' and adults' reports. Comparisons of effect sizes will be made between the arms of the study (control schools and intervention schools). Measures are described in the protocol and in table 2.

Question (4). Will participation in the course Thoughts and Health increase the participants' chances of completing middle school with high school eligibility? The answer will be based on follow-up measurements of completion and final grades at graduation from middle school. Comparisons of effect sizes will be made between the arms of the study (control schools and intervention schools).

## Current trial status

Recruitment of participants started in September 2019, and the last participant is expected to reach the primary endpoint (18-month follow-up) in April 2021. Primary data analysis will begin in May 2021. The naturalistic follow-up phase of the trial will continue until October 2022.

## Patient and public involvement

We will receive input from participants and professionals from focus group discussions and interviews, which will guide the continuation of the intervention study. In the current trial, no participants were involved in the design of the study or in the decision of outcome measures.

Members from the public or students were not involved in designing this study.

We will assess the effect of the trial interventions on the participants by collecting information through e-mailed self-rating questionnaires at the end of programme.

**Author affiliations**
[1]Primary Health Care, School of Public Health and Community Medicine, University of Gothenburg Institute of Medicine, Sahlgrenska Academy, Gothenburg, Sweden
[2]Research and Development Primary Health Care, Region Västra Götaland, Gothenburg, Sweden
[3]Department of Psychology, Psychiatry and Behavioral Sciences, Emory University, Atlanta, Georgia, USA
[4]Faculty of Medicine, School of Health Sciences, Reykjavík, Iceland
[5]Department of Psychiatry, Landspítali-University Hospital, Reykjavík, Iceland
[6]Department of Medicine and Optometry, University of Kalmar, Kalmar, Sweden
[7]Department of Psychology, University of Gothenburg, Goteborg, Sweden

**Acknowledgements** We thank the participating schools and the people who participated in making this project real.Thanks to Maria EH Larsson (Head of Research and Development, primary health care region Västra Götaland.) Thank you school nurses Helena Nordevik Pedersen, Gunilla Börjesson and Inger Höber, psychologists Camilla Johansson, Magnus Hjorth and Henrik Tomasic, head of the primary care centre in Tjörn Ann-Sofie Lekander, GP Torbjörn Erneholm, and head of school health in Tjörn Anna Orvefors, and to Jenna Anderson from Grants Office (Gothia Forum) for help in the funding process.

**Contributors** Substantial contributions to the conception or design of the work: CW, JLL, GS, WEC, IM and EÖA. Planning, conducting and reporting of the work: CW, JLL, PA and GS. Drafting the manuscript: CW and JLL. Critical revision of the article: all declared authors. Final approval of the version published and agreement to be accountable for all aspects of the work: all declared authors.

**Funding** Social investment grants from Region Västra Götaland, Sweden (RUN 2018-00614).

**Competing interests** None declared.

**Patient consent for publication** Not required.

**Provenance and peer review** Not commissioned; externally peer reviewed.

**ORCID iDs**
Carl Wikberg http://orcid.org/0000-0002-6494-5922
Pia Augustsson http://orcid.org/0000-0002-4698-4151
W Edward Craighead http://orcid.org/0000-0001-9957-0027
Josefine L Lilja http://orcid.org/0000-0003-3623-5760

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
