## [Reviewer comments · BMJ Open]

ARTICLE DETAILS

TITLE (PROVISIONAL)	Is the "Thoughts and Health" programme feasible in the context of Swedish schools, a quasi-experimental controlled trial - Study Protocol
AUTHORS	Wikberg, Carl; Augustsson, Pia; Sveinsdottir, Gudny; Craighead, Edward; Arnarson, Erikur; Marteinsdottir, Ina; Lilja, Josefine L.

VERSION 1 – REVIEW

REVIEWER	Prof. Dr. Lence Miloseva Faculty of Medical Sciences, Goce Delcev University-Stip North Macedonia
REVIEW RETURNED	27-Aug-2020

GENERAL COMMENTS	This topic is very important and you are doing great job. It is very important to establish protocol and to offer also to other colleagues to apply it. I would like to emphasis developmental psychology/psychopathology dimension in this protocol which is good point, because it is not so common. The most frequency of the first occurrence of depression is about 15, but there are also subclinical symptoms appearance about 13 years, so the best range according to some my research and my colleagues in the world is from 13 till 17 years. Once you will get data, you can also make predictive model and use different statistical combination (multiple regression analysis) in order to identified the most strong predictive risk factors. Based on them, you will better create personalized preventive programs.
---

REVIEWER	Dr Mairead Cardamone-Breen Monash University, Australia
REVIEW RETURNED	01-Sep-2020

GENERAL COMMENTS	Comments to the authors: This paper presents a study protocol for an uncontrolled evaluation of a CBT-based depression prevention program for adolescents in Sweden. It extends on prior studies of the program and results will add valuable data to a growing field of literature regarding prevention of internalising disorders in adolescents. Despite some limitations acknowledged by the authors, the study design and methodology are appropriate and for the most part well described. My comments below relate to details that could be clarified in the manuscript, rather than limitations of the study protocol. It is noted that many of these suggested changes are requirements of the SPIRIT Checklist. With these revisions, I believe the protocol is appropriate for publication.
--

	Title -Please note that the title does not currently adhere to SPIRIT guidelines re: study design and intervention. Abstract -Please also apply any suggestions below to abstract text, as required. -The abstract would benefit from inclusion of brief information about the intervention itself, as well as the sample/population. Introduction -Overall, the introduction would benefit from a brief explanation of the theory and evidence-base for cognitive-behavioural prevention program for adolescent depression. This seems an important part of the rationale for this study. -Line 27 – please clarify, are these four schools the only four schools in the two municipalities? If not, how were the schools selected? -Line 30 – culturally, are Iceland and Sweden similar, i.e. is there any reason to think that the program would or would not be transferrable to Sweden? A brief comment on this may be helpful, particularly for international audiences. -Lines 33-37 – the control group is mentioned for the first time here, however I am not yet sure what the control condition is. Please make this clear, either here or earlier in the manuscript. -Line 32 – first hypothesis – I believe this should specify comparison to control group. I.e. compared to control, adolescents who receive the intervention will show reduced risk of... -Line 36 – please clarify wording in statement “complete grades to a higher extent” – do you mean the academic grades will be higher, or that more students will complete their schooling? -Line 48 – please clarify the estimated age of ninth graders in Sweden, for international audiences. As per above comment, the control group is mentioned here however I am still not sure what type of control group is being used. Method Study design -Please specify the details of the control condition here. How were the schools allocated to condition? How/why was the choice of control condition selected? -Please ensure you include the required details of trial design as per SPIRIT Checklist requirements. Participants -Line 25 – please specify dates (approximate if needed) of fall semester, as international audiences will not know this. -Was a power analysis conducted, or can the researchers comment on the statistical power given the small sample? Measures and assessments -Page 7, Line 13. Please revise sentence “Outcome Measures Over Time”, unclear what this means. -Page 7, line 22. It is stated that both parents and adolescents will be asked about the parent’s history of depression. Please clarify why both will be asked, and which variable will be used in analyses, e.g. what will be done if reports differ? I would expect there to be discrepancy in report, unsure why adolescent report is being used. -Please clarify who will be conducting the interviews/assessments, and include brief information re: qualifications. -Page 7, line 52 onward – for the questionnaires, please clarify
--	--

whether parent or adolescent or both will complete these.
-Page 8, line 25 – course surveys. Please clarify what specifically these surveys will assess.
-Page 8, line 34 – please clarify who will decide whether participants attend the focus group or individual interview. E.g. is this based on participant preference?
-Please clarify how and when all psychometric measures (particularly survey-based) will be completed. I.e. are these hard copy surveys completed at school, or online surveys sent out to students? If online, please clarify any methods to promote completion/retention over the study period (e.g. reminder protocols). This is particularly relevant for adolescent participants given the longitudinal nature of the study. With such a small sample, it would seem imperative to try to maximise survey completion rates.

Intervention

-Page 9, line 20 – unclear what ‘resume’ refers to in this context, please reword/clarify language.
-It may be helpful to include a table summarising intervention content, e.g. session number, topic, and a little more detail about the content. E.g. from the current text, I am unsure what “setting targets”, “stop and reconsideration” and “consummation” means. This may be an English language issue. The level of detail included for session 2 is very useful, something similar for each session would be great.
-Will intervention fidelity and adherence be assessed and reported on? Please clarify how or explain why not.

Statistical analysis

-Will analyses be intent-to-treat? Given the adolescent sample, I would expect some degree of drop-out/non-completion of outcome measures. It will be good to mention how this will be handled.
-Please provide further details on the qualitative analysis, e.g. instead of “such as thematic analysis”, state that thematic analysis will be used.

Ethics

-Page 9, line 47 – “both parents and the adolescents” – please clarify wording, unclear whether this means both parents (i.e. 2 x parents) or one parent + adolescent.
-For international readers, it may be helpful to briefly mention what the “Good Clinical Practice” is.
-As above, please specify the qualifications/training of the researchers and professionals.

Discussion

-Research Question 1) Will attendance/drop out rates also inform this question? Also ‘course surveys’ referred to in the methods?
-Research Question 2) As above re: surveys administered after each session.
-Research Question 3) Please re-word this paragraph for clarity. Also include the specific measures and how this will be determined.

Tables

Table 1. It would be helpful to clarify the set criteria for the following exclusion criteria: “Not high enough level of function on CGAS” (specify cut-off score); “serious mental illness” (how is this determined?); language (who will assess this)?
-please clarify and ensure consistency throughout paper re: “ongoing psychotherapeutic treatment” earlier in the paper it was

	mentioned that psychological/other professional help would be available to all participants? CONSORT flow diagram – “Anticipated assessed for eligibility” – If I understood correctly, baseline assessments have already been completed, so the actual N could be included here?
--	---

VERSION 1 – AUTHOR RESPONSE

Reviewer: 1

Reviewer Name: Prof. Dr. Lence Miloseva

Institution and Country:

Faculty of Medical Sciences, Goce Delcev University-Stip

North Macedonia

Please state any competing interests or state ‘None declared’: None declared

Please leave your comments for the authors below

This topic is very important and you are doing great job. It is very important to establish protocol and to offer also to other colleagues to apply it.

I would like to emphasis developmental psychology/psychopathology dimension in this protocol which is good point, because it is not so common. The most frequency of the first occurrence of depression is about 15, but there are also subclinical symptoms appearance about 13 years, so the best range according to some my research and my colleagues in the world is from 13 till 17 years. Once you will get data, you can also make predictive model and use different statistical combination (multiple regression analysis) in order to identified the most strong predictive risk factors. Based on them, you will better create personalized preventive programs.

Answer – Thank you for your kind review. We will take your suggestions, and use them in the future once we get more data.

Reviewer: 2

Reviewer Name: Dr Mairead Cardamone-Breen

Institution and Country: Monash University, Australia

Please state any competing interests or state ‘None declared’: None declared

Please leave your comments for the authors below

Comments to the authors:

This paper presents a study protocol for an uncontrolled evaluation of a CBT-based depression prevention program for adolescents in Sweden. It extends on prior studies of the program and results will add valuable data to a growing field of literature regarding prevention of internalising disorders in adolescents. Despite some limitations acknowledged by the authors, the study design and methodology are appropriate and for the most part well described. My comments below relate to details that could be clarified in the manuscript, rather than limitations of the study protocol. It is noted that many of these suggested changes are requirements of the SPIRIT Checklist. With these revisions, I believe the protocol is appropriate for publication.

Title

-Please note that the title does not currently adhere to SPIRIT guidelines re: study design and intervention.

Answer – Thanks for comment, we have made the change.

Abstract

-Please also apply any suggestions below to abstract text, as required.

-The abstract would benefit from inclusion of brief information about the intervention itself, as well as the sample/population.

Answer – We have added information starting on line 87.

Introduction

-Overall, the introduction would benefit from a brief explanation of the theory and evidence-base for cognitive-behavioural prevention program for adolescent depression. This seems an important part of the rationale for this study.

Answer – Thanks for this important comment – We have added a brief explanation starting on page 3 row 126.

-Line 27 – please clarify, are these four schools the only four schools in the two municipalities? If not, how were the schools selected?

Answer – Clarified row 164

-Line 30 – culturally, are Iceland and Sweden similar, i.e. is there any reason to think that the program would or would not be transferrable to Sweden? A brief comment on this may be helpful, particularly for international audiences.

Answer – Added a brief comment on row 167

-Lines 33-37 – the control group is mentioned for the first time here, however I am not yet sure what the control condition is. Please make this clear, either here or earlier in the manuscript.

Answer – Clarified on 158

-Line 32 – first hypothesis – I believe this should specify comparison to control group. I.e. compared to control, adolescents who receive the intervention will show reduced risk of...

Answer – Specified on row 181

-Line 36 – please clarify wording in statement “complete grades to a higher extent” – do you mean the academic grades will be higher, or that more students will complete their schooling?

Answer – Clarified on row 156

-Line 48 – please clarify the estimated age of ninth graders in Sweden, for international audiences. As per above comment, the control group is mentioned here however I am still not sure what type of control group is being used.

Answer – Clarified age on row 181

Method

Study design

-Please specify the details of the control condition here. How were the schools allocated to condition? How/why was the choice of control condition selected?

Answer – We have specified this starting on row 200.

-Please ensure you include the required details of trial design as per SPIRIT Checklist requirements.

Answer - Corrected

Participants

-Line 25 – please specify dates (approximate if needed) of fall semester, as international audiences will not know this.

Answer – We have specified dates row 207

-Was a power analysis conducted, or can the researchers comment on the statistical power given the small sample?

Answer – Added explanation on 215

Measures and assessments

-Page 7, Line 13. Please revise sentence “Outcome Measures Over Time”, unclear what this means.
Answer – Thanks for comment we have removed that sentence.

-Page 7, line 22. It is stated that both parents and adolescents will be asked about the parent's history of depression. Please clarify why both will be asked, and which variable will be used in analyses, e.g. what will be done if reports differ? I would expect there to be discrepancy in report, unsure why adolescent report is being used.

Answer – We have added sentences elaborating on the issue. row 309

-Please clarify who will be conducting the interviews/assessments, and include brief information re: qualifications.

Answer – We have clarified this on row 332

-Page 7, line 52 onward – for the questionnaires, please clarify whether parent or adolescent or both will complete these.

Answer – This information is updated and available in table 2.

-Page 8, line 25 – course surveys. Please clarify what specifically these surveys will assess.

Answer – We have clarified row 316

-Page 8, line 34 – please clarify who will decide whether participants attend the focus group or individual interview. E.g. is this based on participant preference?

Answer – Clarified on row 325

-Please clarify how and when all psychometric measures (particularly survey-based) will be completed. I.e. are these hard copy surveys completed at school, or online surveys sent out to students? If online, please clarify any methods to promote completion/retention over the study period (e.g. reminder protocols). This is particularly relevant for adolescent participants given the longitudinal nature of the study. With such a small sample, it would seem imperative to try to maximise survey completion rates.

Answer – Clarified on row 245.

Intervention

-Page 9, line 20 – unclear what ‘resume’ refers to in this context, please reword/clarify language.

Answer – Added clarification row 358

-It may be helpful to include a table summarising intervention content, e.g. session number, topic, and a little more detail about the content. E.g. from the current text, I am unsure what “setting targets”, “stop and reconsideration” and “consummation” means. This may be an English language issue. The level of detail included for session 2 is very useful, something similar for each session would be great.

Answer – Thanks for the comment, we believe the level of detail is enough to get an idea of the content. We are willing to add a full manual of the program describing each session in thorough details as an attachment if needed.

-Will intervention fidelity and adherence be assessed and reported on? Please clarify how or explain why not.

Answer – In this study, we have other aims. We are first interested in if the intervention is feasible in Sweden then the possible effects it may have on preventing depression and effects on school results.

Statistical analysis

-Will analyses be intent-to-treat? Given the adolescent sample, I would expect some degree of drop-out/non-completion of outcome measures. It will be good to mention how this will be handled.

-Please provide further details on the qualitative analysis, e.g. instead of “such as thematic analysis”, state that thematic analysis will be used.

Answer – Thanks for this comment. We have now added some clarification on row 373.

Ethics

-Page 9, line 47 – “both parents and the adolescents” – please clarify wording, unclear whether this means both parents (i.e. 2 x parents) or one parent + adolescent.

Answer – Clarified on row 381

-For international readers, it may be helpful to briefly mention what the “Good Clinical Practice” is.

Answer – Done. Row 396.

-As above, please specify the qualifications/training of the researchers and professionals.

Answer – Done. Row 400

Discussion

-Research Question 1) Will attendance/drop out rates also inform this question? Also ‘course surveys’ referred to in the methods?

Answer –Added on row 421

-Research Question 2) As above re: surveys administered after each session.

Answer – Added row 427

-Research Question 3) Please re-word this paragraph for clarity. Also include the specific measures and how this will be determined.

Answer – Added clarification row 428

Tables

Table 1. It would be helpful to clarify the set criteria for the following exclusion criteria: “Not high enough level of function on CGAS” (specify cut-off score); “serious mental illness” (how is this determined?); language (who will assess this)?

Answer – Clarified.

-please clarify and ensure consistency throughout paper re: “ongoing psychotherapeutic treatment” earlier in the paper it was mentioned that psychological/other professional help would be available to all participants?

Answer – Clarified

CONSORT flow diagram – “Anticipated assessed for eligibility” – If I understood correctly, baseline assessments have already been completed, so the actual N could be included here?

Answer – Corrected. See attachment!

VERSION 2 – REVIEW

REVIEWER	Dr Mairead Cardamone-Breen Monash University, Australia
REVIEW RETURNED	11-Nov-2020

GENERAL COMMENTS	Thank you for the opportunity to review a revised version of this manuscript. The authors have made most of the requested changes to a satisfactory level. Please note that the line/page numbers noted in the responses to review do not match either of the manuscript versions provided, which made it difficult to assess each change against the comments. Nonetheless, my only remaining minor concern is the English language expression. The paper would benefit from careful proofreading for spelling, grammar, and consistency of language. I believe this can be addressed during copyediting. I look forward to reading the results of this promising study.
--

VERSION 2 – AUTHOR RESPONSE

We are grateful for the comments and have added the requested inquiries from the editor and reviewer to the manuscriptcentral.

We will answer each request below.

- Please provide an English version of the model consent form and ensure that this is made available for review when it's uploaded as supplementary file.

Answer - We have added a English version of the consent

- Please amend the meta-data in the submission system to reflect the fact that your proposed study IS a clinical trial.

Answer - Done

- To adhere to ICMJE guidelines, we require that a data sharing plan must be included with trial registration for clinical trials that begin enrolling participants on or after 1st January 2019. Please update your trial registry entry with this information.

Answer - Done in clinicaltrials.com

- Please work to improve the quality of the English throughout your manuscript. We recommend asking a native English speaking colleague to assist you or to enlist the help of a professional copy-editing service.

Answer - We have hired a professional copy-editor to improve the language

We hope the adjustments are satisfying and that the manuscript now will be accepted for publication.